# Effects of boysenberry on postprandial energy metabolism in healthy adults: A randomized controlled crossover trial

Ryo Furuuchi[1,2*], Satoshi Kato[1], Daisuke Maejima[1], Tatsuro Amano[3], Ippei Shimizu[4], Tohru Minamino[5]

1 Advanced Research Institutes, Bourbon Corporation, Niigata, Japan, 2 Department of Advanced Senotherapeutics, Juntendo University Graduate School of Medicine, Tokyo, Japan, 3 Laboratory for Exercise and Environment Physiology, Faculty of Education, Niigata University, Niigata, Japan, 4 Department of Cardiovascular Aging, National Cerebral and Cardiovascular Center Research Institute, Osaka, Japan, 5 Department of Cardiovascular Biology and Medicine, Juntendo University Graduate School of Medicine, Tokyo, Japan

* furuuchi-ryo@bourbon.co.jp

## Abstract

Brown adipose tissue (BAT) is essential for thermoregulation and energy metabolism, converting fatty acids into thermal energy in response to cold exposure and dietary intake, thereby contributing to both cold-induced thermogenesis and diet-induced thermogenesis (DIT). Our previous research suggests that boysenberry anthocyanins (BoyACs) may activate BAT under cold conditions, and we hypothesized that BoyACs could also influence DIT through the activation of BAT. This pilot randomized, double-blind crossover trial aimed to evaluate the effects of daily intake of BoyACs on DIT in healthy adults (registration number: UMIN000047413). Twenty-two participants consumed either a boysenberry juice (BoyJ) beverage containing 61.0 mg of BoyACs or a placebo beverage daily for four weeks, with a four-week washout period separating the two interventions. Three participants withdrew during the trial, resulting in data from 19 participants being analyzed. Results showed no significant changes in DIT, defined as increase in postprandial energy expenditure, or skin temperature of BAT regions. However, a significant increase in postprandial fat oxidation was observed. No significant differences were observed in other outcomes. These results suggest that BoyJ intake does not significantly affect postprandial energy expenditure but may influence substrate utilization to promote fat oxidation. Further studies focusing on substrate utilization, particularly fat oxidation, as the primary outcome are necessary to confirm these results and fully understand the implications of BoyJ intake on energy metabolism.

**Data availability statement:** All relevant data are within the manuscript and its Supporting Information files.

**Funding:** The author(s) received no specific funding for this work.

**Competing interests:** R.F., S.K., and D.M. are affiliated with Bourbon Corporation as researchers. BoyJ and placebo were supplied by Bourbon Corporation for the study. This study was conducted with joint research funds provided by Bourbon Corporation. Bourbon Corporation and Juntendo University have established a collaborative research program funded by Bourbon Corporation, and the program is organized by T.M. The other authors have no conflicts of interest to disclose.

## Introduction

Brown adipose tissue (BAT) is known as thermogenic tissue and plays an important role in thermoregulation under cold conditions [1]. When activated by cold stimuli, BAT is efficiently converted into thermal energy using fatty acids as substrates. BAT is thought to increase energy metabolism and prevent excess fat accumulation through these mechanisms [2].

In addition to its role in cold-induced thermogenesis, BAT has been reported to be associated with diet-induced thermogenesis (DIT) [3,4]. DIT occurs during the digestion, absorption, metabolism, and storage of ingested food and is estimated to account for 5% to 15% of overall energy consumption in daily life [5]. For this reason, DIT is considered to play an important role in weight management and obesity prevention. Thus, activation of BAT may promote fat oxidation by increasing energy expenditure (EE) via DIT, which may contribute to weight loss and prevention of metabolic diseases.

Recent findings from preclinical studies suggest that boysenberry anthocyanins (BoyACs) have a beneficial effect by suppressing the decline in BAT function in models of BAT dysfunction. In fact, it was suggested that BoyACs may significantly decrease the respiratory quotient (RQ) during the active phase of mice with impaired BAT function, leading to increased fat oxidation and EE [6]. Additionally, results of a preliminary clinical trial suggest that intake of boysenberry juice (BoyJ) induces BAT thermogenesis in humans during cold exposure [7]. The mechanism by which BoyACs affect BAT function is thought to involve activation of sirtuin 1, which is involved in activation of metabolic pathways that enhance fat oxidation [6], similar to the effects observed with certain polyphenols known to activate sirtuin 1 [8]. Based on this, we hypothesized that boysenberries may influence DIT and associated energy metabolism in humans through the activation of BAT.

To evaluate the effect of BoyACs on DIT and energy metabolism, we conducted a pilot study to serve as a foundation for future larger trials. The study was designed as a randomized, double-blind crossover comparative trial in which participants consumed BoyJ containing BoyACs for four weeks.

## Methods

### Study design

We used a randomized, double-blind crossover study design. The study was approved by the Ethics Review Committee of the Juntendo University School of Medicine (E21-0360-M01, 18/3/2022) and was conducted in accordance with the Declaration of Helsinki. The study content was registered in the UMIN Clinical Trials Registry (registration number: UMIN000047413, 11/04/2022). Participants were male and female volunteers aged 20 years or older with a BMI between 18.5 and 25.0 kg/m$^2$. Exclusion criteria included illnesses, allergies, medication or health supplement use, polyphenol-rich diets, planned lifestyle changes, and pregnancy. Participants were recruited in Niigata, Japan, between 10/4/2022 and 31/5/2022. The trial was initiated on 4/6/2022 and completed on 3/10/2022, during which participants received

the intervention. The data were collected at Niigata University. Informed consent was obtained from all participants prior to participation in the study.

There were 30 participants willing to participate in the study. Participants provided informed consent in writing. After screening, 7 participants were excluded due to chronic disease, medication intake, BMI < 18, and age, leaving 23 participants included in the study. Out of the 23 participants, 1 withdrew for personal reasons prior to randomization, and the remaining 22 were randomized into two groups. During the first intake period, 2 participants became infected with COVID-19 and 1 withdrew for personal reasons. As a result, 19 participants completed the study. Three dropouts were excluded from the statistical analysis because all outcome data were not obtained, and the Full Analysis Set was performed on 19 participants. The study flow diagram is shown in Fig 1.

### Test foods

The test food was BoyJ beverage (Bourbon Corporation). The placebo was a drink that was indistinguishable from BoyJ in taste and appearance. Participants consumed 100 ml of either BoyJ or placebo daily for 4 weeks; the time of consumption was not specified.

BoyAC content was determined by high-performance liquid chromatography [7]. The four detected components (cyanidin-3-2-glucosylglucoside, cyanidin-3-2-glucosylrhamnosylglucoside, cyanidin-3-glucoside, and cyanidin-3-6- rhamnosylglucoside) were calculated as equivalents of cyanidin-3-glucoside (Nagara Science Corporation, code: 639-43451). The total amount was defined as the amount of BoyACs, which was 61 mg per 100 ml of BoyJ (Table 1). The placebo contained no BoyACs.

### Intervention

The crossover design included a 4-week intake period (Phase I), a 4-week washout period, and another 4-week intake period (Phase II) from June to September 2022. The primary outcome measure for evaluating DIT was calculated from the increase in postprandial EE. Additionally, time-course data of EE changes and skin temperature in the BAT region were evaluated to assess DIT. Secondary outcomes were changes in BMI and postprandial hand skin temperature, oxygen consumption, carbohydrate oxidation, fat oxidation, respiratory quotient, appetite, and cold sensation. Participants were randomly assigned to test foods using simple randomization, with an allocation list created by a person not involved in the study. The list was kept sealed and securely stored to maintain blinding for the research investigator, participants, measurement staff, and all other personnel involved in the study until the completion of the trial.

Participants consumed either BoyJ or a placebo drink containing no BoyACs daily for four weeks. After 4 weeks of intake, participants visited the study site for testing. Participants were instructed to refrain from exercise the day before testing, to eat the same dinner before the first and second test days, and to refrain from eating or drinking anything other than water after 9:00 p.m. the day before testing. At the test site of the Niigata University research facility, participants changed into a tank top, shorts, and socks and rested in a room at 27 ± 1°C for 30 minutes. After rest, baseline (0 min) skin temperature, respiratory gas, and Visual Analog Scale (VAS) measurements were taken. Participants consumed 200 ml of water and a meal (CalorieMate, Otsuka Pharmaceutical Co., Ltd.) that provided 15% of their total EE based on the Harris-Benedict Calculator. CalorieMate contains 500 kcal of energy per 100 g, consisting of 10.5 g of protein, 27.8 g of fat, 53.4 g of carbohydrates, and 1.2 g of salt. Skin temperature, respiratory gas, and VAS measurements were taken 30, 60, 90, 120, 150, and 180 minutes after the start of the meal (Fig 2).

### Respiratory gas analysis

To assess energy metabolism, respiratory gas was analyzed using the Douglas bag method. The concentrations of oxygen and carbon dioxide in the expired air collected in the Douglas bag were analyzed using a gas analyzer (Off-gas Jr,

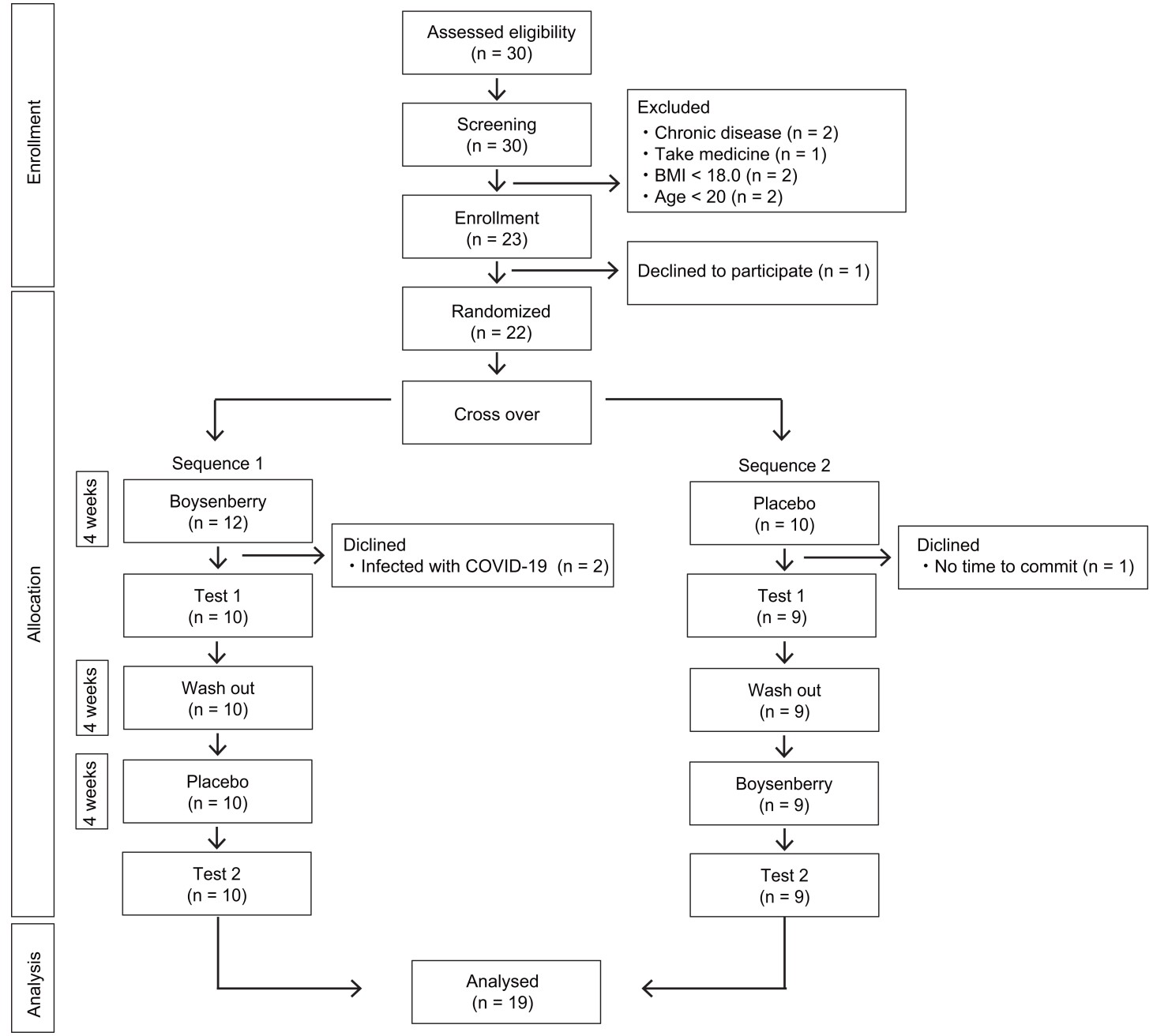

**Fig 1. Study flow diagram.** A total of 30 participants expressed interest in the study. Out of these, 22 were randomized to participate in the crossover trial. Three participants dropped out, resulting in a final analysis of 19 participants.

Able). The volume and temperature of the expired gas were measured using a gas meter (DC-5A, Shinagawa). EE, fat oxidation, and carbohydrate oxidation were calculated using the following formulas [9,10]:

$$EE\ (kcal/min) = (3.9 \times VO_2 + 1.1 \times VCO_2) \times 1.44$$

**Table 1. Nutrients in boysenberry juice (BoyJ) and placebo.**

| | BoyJ (100 g) | Placebo (100 g) |
|---|---|---|
| Energy content, kcal | 32.0 | 32.0 |
| Protein, g | 0.5 | 0.0 |
| Fat, g | < 0.1 | 0.0 |
| Carbohydrate, g | 7.2 | 7.9 |
| Ellagic acid, mg | 8.0 | 0.0 |
| Anthocyanins[a] | | |
| Cyanidin-3–2-glucosylglucoside, mg | 27.6 | 0.0 |
| Cyanidin-3–2-glucosylrhamnosylglucoside, mg | 18.5 | 0.0 |
| Cyanidin-3-glucoside, mg | 12.8 | 0.0 |
| Cyanidin-3–6- rhamnosylglucoside, mg | 2.1 | 0.0 |
| Total anthocyanins (BoyACs), mg | 61.0 | 0.0 |

[a]Quantitative values were calculated as cyanidin-3-glucoside equivalents.

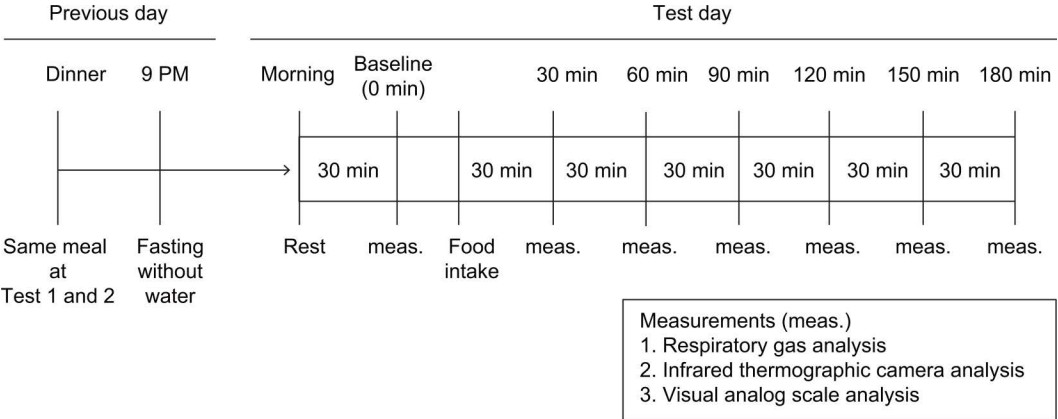

**Fig 2. Test schedule for the study.** Participants were instructed to consume the same meal and to fast after 9:00 p.m. before both test days. Participants rested for 30 minutes in the testing area, then baseline measurements were taken (0 min) before they consumed the meal. Measurements were taken every 30 minutes for a total of 180 minutes, starting from the beginning of the meal.

$$\text{Carbohydrate oxidation (mg/min)} = (4.55 \times \text{VCO}_2 - 3.21 \times \text{VO}_2) \times 1000$$

$$\text{Fat oxidation (mg/min)} = (1.67 \times \text{VO}_2 - 1.67 \times \text{VCO}_2) \times 1000$$

$$\text{Respiratory quotient (RQ)} = \text{VCO}_2/\text{VO}_2$$

RQ values >1.0 result in negative fat oxidation values, which were corrected to zero [11]. DIT was calculated by determining the area under the curve for the increase in postprandial EE and dividing it by the caloric intake [5].

## Skin surface temperature

Skin surface temperatures were measured with an infrared thermographic camera (Testo 885, Testo K.K. Co., Ltd.). Average temperatures of the supraclavicular region (Tscv), upper chest (Tch), and backs of the hands from wrist to fingertips (Thand) were determined. BAT activity was assessed by calculating the difference (Tscv−ch) between Tch (an area of low BAT activity) and Tscv (an area of high BAT activity) [12].

## VAS

We assessed participants' perception of coldness and motivation to eat using VASs. Each VAS was a 100 mm line on which participants placed a pencil mark to describe their feelings along the continuum. Participants rated their perception of coldness from "not cold" to "very cold". For motivation to eat, participants rated their desire to eat ("very weak" to "very strong"), hunger ("not hungry at all" to "as hungry as I have ever felt"), fullness ("not full at all" to "very full"), and prospective food consumption (PFC, "a large amount" to "nothing at all"). Based on these ratings, a subjective appetite score was calculated using the following formula [13]:

$$\text{Appetite (mm)} = [\text{desire to eat} + \text{hunger} + (100 - \text{fullness}) + \text{PFC}]/4$$

## Statistical analysis

The target sample size was determined based on a preliminary trial we conducted, using the increase in Tscv during cold exposure before and after test meal ingestion as an indicator of BAT activity. In previous literature [14], 21 male participants were evaluated for DIT and 23 male participants for cold induced thermogenesis (CIT), with both groups showing a significant increase in high BAT activity. This suggests that similar effects may be expected between DIT and CIT. However, due to the differences in the metrics, the analysis was conducted with a power calculation of 95%. With a statistical power of 95% and a significance level of 0.05, the necessary sample size was determined to be 22 participants. However, because the present study used different indicators than the preliminary trial, we also referenced prior literature [15], which showed significant differences in a crossover trial with 36 participants. Consequently, the target sample size for the present study was set at 22–36 participants.

Results are presented as mean ± standard error (SE) or CI. Fasting measurements were compared using paired t-tests. For each measurement parameter, the difference between the baseline (0 min) and post-meal measurements was calculated to determine the changes induced by the meal, and statistical analyses were performed on these differences. DIT was compared using a paired t-test. Comparisons of change over time between groups were conducted using linear mixed- models. The fixed effects in the model included period, time, order, and measurement value. Participant ID was included as a random effect to account for inter-individual variability. Repeated measures over time were modeled using a first-order autoregressive correlation. To evaluate the potential carryover effect, the sum of DIT (%) for Test 1 and Test 2 was calculated, and the sequences 1 and 2 were compared using a Student's t-test. To consider the sex balance across sequences, we analyzed the effects using the chi-squared test. The significance level for all tests was set at 0.05. Statistical analyses were performed using IBM SPSS Statistics version 28.0.1. The validity of the crossover study was confirmed, as there were no significant order or timing effects (data not shown).

## Results

### Baseline characteristics

Baseline and demographic characteristics of the 19 participants are shown in Table 2. The participants included 12 men and 7 women, with a mean age of 34.5 ± 3.5 years and a mean BMI of 21.8 ± 0.3 kg/m². During the trial period, one

**Table 2. Characteristics of participants before and after beverage intake.**

|  | Pre | Placebo | BoyJ | *P* value |
|---|---|---|---|---|
| Male to female ratio | 12:7 | – | – | – |
| Age | 34.5±3.5 | – | – | – |
| Height, cm | 167.0±1.9 | 167.3±2.0 | 167.3±2.0 | 0.674 |
| Body weight, kg | 60.8±1.7 | 60.5±1.6 | 60.4±1.6 | 0.608 |
| BMI[a], kg/m$^2$ | 21.8±0.3 | 21.6±0.3 | 21.5±0.3 | 0.819 |

Data represent the mean±SEM. Comparisons between placebo and BoyJ intake were performed using paired t-tests. [a] Body mass index (BMI) is weight (kilograms) divided by height (meters) squared. The source data are shown in S1 Table.

participant dropped out before randomization, and three participants dropped out after randomization, which hindered our ability to collect the target sample size. We confirmed that there was no significant difference in the number of males and females across sequences (Sequence 1: 7 males and 3 females; Sequence 2: 5 males and 4 females), with a p-value of 0.515.

No significant differences were observed between the BoyJ and placebo groups in the baseline (0 min) data during the fasting period (Table 3). No significant health harms or unintended effects were observed in any participants throughout the study.

## BoyJ does not promote DIT

The paired t-test showed no significant changes between the BoyJ and placebo groups in the primary outcome measure, DIT (Fig 3A, *p*=0.331). In addition, linear mixed-effects models showed no significant differences in the time-course data of postprandial EE changes (Fig 3A, *p*=0.487). BAT activity was evaluated by calculating the temperature difference between the upper chest (Tch) and the supraclavicular region (Tscv) [12,16]. There was no significant difference in ΔTscv−ch after the meal between the placebo and BoyJ groups (Fig 3C, *p*=0.133), as determined by linear mixed-effects models. These results suggest that BoyJ did not affect BAT-mediated DIT in this experiment. The results of the primary and secondary outcomes are summarized in Table 4. Additionally, no carryover effect was observed for DIT (p=0.595).

## BoyJ increases postprandial fat oxidation

As a secondary outcome, we evaluated the effect of BoyJ intake on postprandial substrate metabolism. Postprandial fat oxidation (Δfat oxidation) was significantly higher in the BoyJ group than the placebo group, with an estimated intervention effect of +5.60 mg/min (95% CI, 0.88 to 10.31, *p*=0.021), as illustrated in Fig 4A. Postprandial carbohydrate oxidation (Δcarbohydrate oxidation, Fig 4B, *p*=0.587), postprandial oxygen consumption (ΔVO2, Fig 4C, *p*=0.437), respiratory quotient (ΔRQ, Fig 4D, *p*=0.935) did not differ significantly between the BoyJ and placebo groups. These results suggest that BoyJ increases fat oxidation in postprandial substrate utilization. Changes in appetite associated with DIT and substrate utilization were assessed using a VAS but did not differ significantly between intervention groups (Fig 4E, *p*=0.343).

## Effect on temperature perception

To evaluate whether changes in temperature perception are associated with DIT, cold sensation was assessed with a VAS and hand skin temperature (ΔThand) was measured. There were no significant differences in cold sensation (Fig 4F, *p*=0.341) or ΔThand (Fig 4G, *p*=0.509) between the BoyJ and placebo groups, indicating that there was no significant difference in real or perceived body warmth.

**Table 3. Baseline measurements after 4 weeks' intake of placebo or boysenberry juice (BoyJ).**

|  | Placebo | BoyJ | P value |
|---|---|---|---|
| Energy expenditure, kcal/min | 0.919±0.047 | 0.881±0.064 | 0.524 |
| Fat oxidation, mg/min | 43.7±5.8 | 41.6±6.8 | 0.806 |
| Carbohydrate oxidation, mg/min | 141.2±15.8 | 133.5±14.6 | 0.724 |
| VO$_2$, L/min | 0.186±0.009 | 0.182±0.014 | 0.524 |
| RQ[a] | 0.878±0.024 | 0.877±0.020 | 0.979 |
| Tch[b], °C | 34.4±0.1 | 34.3±0.1 | 0.456 |
| Tscv[c], °C | 35.4±0.1 | 35.4±0.1 | 0.801 |
| Tscv−ch[d], °C | 1.06±0.08 | 1.13±0.06 | 0.241 |
| Thand[e], °C | 34.7±0.2 | 34.5±0.3 | 0.606 |

Data represent the mean±SEM. Comparisons between placebo and BoyJ intake were performed using paired t-tests. [a] Respiratory quotient. [b] Skin surface temperature of the upper chest. [c] Skin surface temperature of the supraclavicular. [d] Difference between Tscv and Tch. [e] Skin surface temperature of the hand. The source data are shown in S2 Table.

## Discussion

This randomized, double-blind crossover study evaluated the effect of BoyAC-containing BoyJ on DIT in healthy men and women aged 20 years and older. Recent studies have shown that BAT promotes energy metabolism via DIT [3,4], which has attracted attention for its potential role in supporting metabolic health, regulating weight, and enhancing overall physiological function. In this study, we found that BoyJ did not increase DIT or skin temperature in the scv region, a marker of BAT activation. However, we found that 4 weeks of BoyJ consumption increased postprandial fat oxidation, suggesting that BoyJ may influence substrate selection in energy metabolism.

A previous study [17] have provided evidence that the consumption of the anthocyanin cyanidin-3-glucoside results in the detection of anthocyanin-derived metabolites in humans for up to 48 hours. They reported that the half-life of these metabolites, including other components, ranged from 12.44±4.22 hours to 51.62±22.55 hours. Therefore, it is considered that the carryover of metabolites after four weeks has minimal impact. Additionally, as the carryover effect was statistically negated in this study, a 4-week washout period was likely sufficient.

BoyAC has been suggested to activate BAT [6,7], leading us to hypothesize that continued BoyAC intake may increase DIT through BAT activation. Contrary to this hypothesis, our findings did not demonstrate that BoyJ consumption affected DIT or thermogenesis in the scv region. Thus, these results suggest that four weeks of BoyJ intake may not affect BAT-mediated DIT. However, limitations in the study's measurement methods and sample size may have limited the ability to detect changes in DIT. In our analysis, the post hoc power calculation yielded a value of 0.250 for DIT, indicating that the study was underpowered to detect an effect of the indicated size. The calculated effect size $d$ was 0.311, suggesting a small effect was present. However, the low power highlights that the results may be considered inconclusive due to underpowering. Therefore, we recommend that future studies with larger sample sizes be conducted to more definitively assess the impact. BAT activity is typically assessed by fluorodeoxyglucose positron emission tomography/computed tomography or thermographic imaging of the scv region under cold-loading conditions.[18,19] Our previous studies have suggested that continuous ingestion of BoyJ containing BoyACs may enhance thermogenesis in the scv region and activate BAT under cold-loading conditions [7]. Since we did not utilize cold-loading conditions in this study, the measurement method may have lacked the sensitivity to detect subtle changes in DIT.

Consistent with previous research on other anthocyanin-rich berries, our study showed that continuous consumption of BoyJ may significantly enhance fat oxidation, leading to preferential selection of fat as a substrate for energy metabolism. After 4 weeks of BoyJ consumption, we observed a significant increase in postprandial fat oxidation. This finding aligns with reports that anthocyanins in blackcurrants [20] and blackberries [21] alter the balance of substrate consumption,

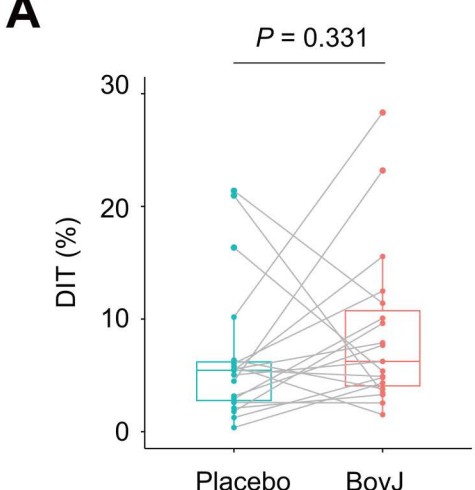

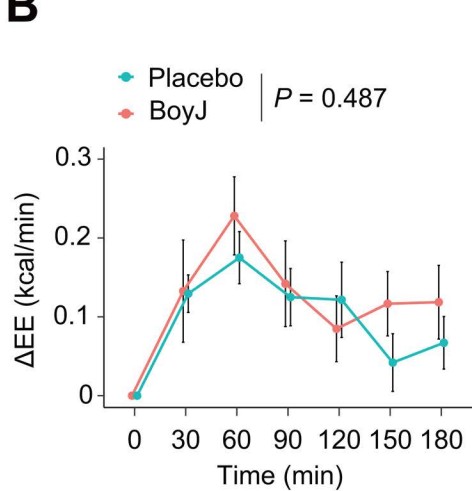

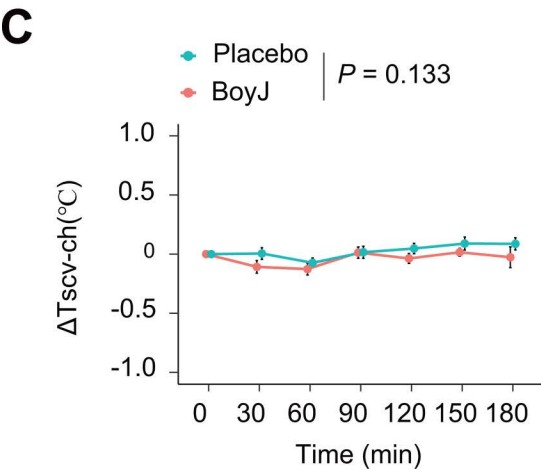

**Fig 3. Effect of boysenberry juice (BoyJ) intake on diet-induced thermogenesis (DIT).** (A) DIT was determined by calculating the area under the curve (AUC) for the increase in postprandial energy expenditure (EE) following the consumption of boysenberry juice (BoyJ) and dividing it by the caloric intake from the meal. (B) Changes in postprandial energy expenditure (ΔEE). (C) The differences in skin surface temperature between the supraclavicular region and chest (ΔTscv−ch). The data are expressed as mean±SE. Comparisons between placebo and BoyJ intake were performed using a paired t-test for A), while B) and C) were analyzed using linear mixed-effects models. *P* values are shown in the figure. The source data are shown in S3 Table, which contains the data for DIT, while S4 Table includes the remaining data.

increasing fat oxidation without elevating EE during exercise in humans. Our study did not find significant differences in EE, RQ, or carbohydrate oxidation. Considering the balance of energy metabolism, this result may seem contradictory. In studies investigating the effects of anthocyanins on human energy metabolism, the reported intake levels of elderberry [22] and blackberry [21] are 755 mg/day and 361 mg/day, respectively, which are significantly higher than the 61 mg/day consumed in our study. Remarkably, our research demonstrated that fat oxidation is promoted even at this relatively low dose of just 61 mg/day, suggesting a potent efficacy of boysenberry. On the other hand, the absence of significant

**Table 4. Results of primary and secondary outcomes.**

| Outcome | Estimated intervention effect | 95% CI[e] | | P value |
|---|---|---|---|---|
| | | Lower | Upper | |
| DIT (%) | 2.07 | −2.28 | 6.41 | 0.331 |
| ΔEE[a], kcal/min | 0.02 | −0.04 | 0.08 | 0.487 |
| ΔFat oxidation, mg/min | 5.60 | 0.88 | 10.31 | 0.021* |
| ΔCarbohydrate oxidation, mg/min | −5.26 | −24.59 | 14.08 | 0.587 |
| ΔVO$_2$, L/min | 0.08 | −0.12 | 0.28 | 0.437 |
| ΔRQ[b] | 0.00 | −0.01 | 0.02 | 0.935 |
| ΔTscv-ch[c], °C | −0.05 | −0.12 | 0.02 | 0.133 |
| ΔThand[d], °C | 0.11 | −0.23 | 0.46 | 0.509 |
| ΔAppetite, mm | 3.20 | −3.58 | 9.98 | 0.343 |
| ΔCold sensation, mm | 0.43 | −0.47 | 1.32 | 0.341 |

Δ indicates the change in values from 0 min. Comparisons between placebo and boysenberry juice (BoyJ) intake were performed using a paired t-test for diet-induced thermogenesis (DIT) and linear mixed-effects models for other outcomes. [a]Energy expenditure. [b]Respiratory quotient. [c]Difference between skin surface temperature of the supraclavicular (Tscv) and the upper chest (Tch). [d]Skin surface temperature of the hand. [e]95% confidence interval (CI). *$P < 0.05$.

differences in carbohydrate oxidation and RQ may indicate that dosage-related issues impacted measurement sensitivity. A study on blackcurrant anthocyanins [20] examined the dose-dependent effects on fat oxidation during exercise. This study found that a moderate intake of 105 mg/day had limited effects compared to higher doses, suggesting the existence of an intake threshold. We acknowledge that these considerations regarding dosage may have influenced the sensitivity of our measurements in the study.

Furthermore, protein oxidation could be an important factor depending on the circumstances, but it was not measured in this study. It is essential to note that DIT is not solely derived from BAT activation, but other physiological processes, including protein synthesis, substrate transport, and peristalsis, also contribute to DIT and play a crucial role in the overall metabolic response to food intake [5]. AMP-activated protein kinase (AMPK) and sirtuin 1, as metabolic sensors, regulate the activity of the master regulator of mitochondria, Peroxisome proliferator-activated receptor gamma coactivator-1-alpha (PGC-1α), and they interact to control energy metabolism in the liver, muscle, and adipose tissue, playing a crucial role in maintaining energy homeostasis [23]. Furthermore, it has been reported that the stimulation of sirtuin 1, AMPK, and PGC-1α, along with their antioxidant effects, enables anthocyanins to promote mitochondrial density and function in skeletal muscle, liver, and adipose tissue [24]. BoyAC has been suggested in preclinical studies to indirectly maintain BAT activation by preserving vascular homeostasis through sirtuin 1 [6]. This effect may influence not only BAT but also a wide range of organs. As a result, while BoyJ appears to promote fat oxidation, we cannot fully determine the relationship between energy metabolism pathways and EE. Thus, further investigation may be needed to elucidate the interactions between anthocyanins, fat oxidation, and relevant tissues, as well as the metabolic pathways involved in these processes.

Additionally, appetite was not significantly affected by BoyJ intake, indicating that secondary effects of increased fat oxidation, such as increased food intake, are unlikely.

## Conclusion

In this pilot study, we evaluated the effects of continuous intake of BoyJ with BoyAC on postprandial thermogenesis and energy metabolism. Although no significant changes were observed in DIT or thermogenesis in the BAT area, the low power of the study limited our ability to draw definitive conclusions regarding DIT. BoyJ intake significantly increased postprandial fat oxidation. These results suggest that continuous consumption of BoyJ may shift energy utilization toward fat

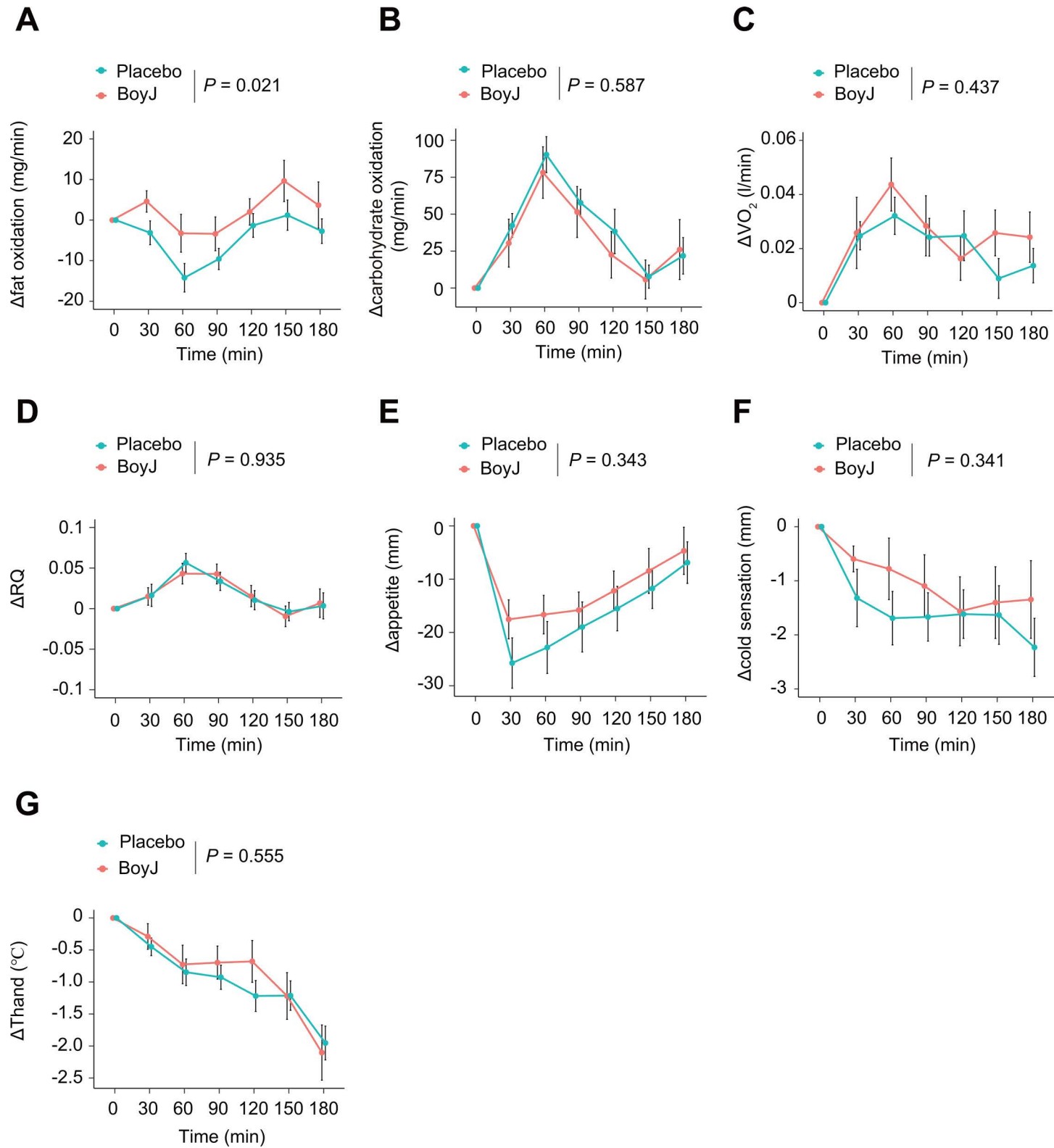

**Fig 4. Effects of boysenberry juice (BoyJ) intake on postprandial changes in energy metabolism, appetite, and temperature perception.** (A) Changes in postprandial fat oxidation (Δfat oxidation), (B) changes in postprandial carbohydrate oxidation (Δcarbohydrate oxidation), (C) changes in

postprandial VO$_2$ (ΔVO$_2$), and (D) changes in postprandial respiratory quotient (ΔRQ) were determined by respiratory gas analysis. (E) Changes in postprandial appetite (Δappetite) and (F) cold sensation were measured using a VAS. (G) Changes in postprandial hand temperature (ΔThand) were measured by infrared thermographic camera. The data are expressed as mean±SE. Comparisons between placebo and BoyJ intake were performed using linear mixed-effects models. $P$ values are shown in the figure. * $P < 0.05$. The source data are shown in S4 Table.

oxidation and promote fat metabolism. However, because this trial was conducted in healthy adults, it is unclear whether these effects would be beneficial for individuals with metabolic syndrome. Future investigations of that population are needed to confirm these findings and assess their implications for metabolic health. Furthermore, due to the limited sample size, further validation studies are necessary to confirm these results.

## Limitations

A significant limitation of the study is that it was conducted during the peak of the COVID-19 pandemic, which hindered our ability to recruit the predetermined number of participants. As a result, the study may have lacked sufficient statistical power to detect true effects. Furthermore, we cannot rule out the possibility that the observed difference in fat oxidation occurred by chance due to statistical multiplicity caused by the inclusion of multiple secondary outcomes.

## Supporting information

**S1 Table. Individual participant characteristics.**
(XLSX)

**S2 Table. Baseline measurements for each participant.**
(XLSX)

**S3 Table. Diet-induced thermogenesis (DIT) for each participant.**
(XLSX)

**S4 Table. Energy metabolism, visual analog scale measurements, and skin surface temperature for each participant.**
(XLSX)

**S1 File. CONSORT 2010 checklist.**
(DOCX)

**S2 File. Research_protocol (ENG).**
(PDF)

**S3 File. Research_protocol (JPN).**
(PDF)

## Acknowledgments

We would like to thank Dr. Saki Uchiyama of the Juntendo University Hospital Clinical Research and Clinical Trial Center for providing advice on the statistical analysis methods and their interpretation.

## Author contributions

**Conceptualization:** Ryo Furuuchi.

**Data curation:** Tatsuro Amano.

**Funding acquisition:** Daisuke Maejima.

**Investigation:** Satoshi Kato, Tatsuro Amano.

**Resources:** Satoshi Kato, Daisuke Maejima.

**Supervision:** Ippei Shimizu, Tohru Minamino.

**Writing – original draft:** Ryo Furuuchi, Daisuke Maejima.

**Writing – review & editing:** Ryo Furuuchi, Tohru Minamino.

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
