## [Decision Letter · Decision Letter 0]

13 Jun 2025

PONE-D-25-23279Effects of Boysenberry on Postprandial Energy Metabolism in Healthy Adults: A Randomized Controlled Crossover TrialPLOS ONE

Dear Dr. Furuuchi,

Thank you for submitting your manuscript to PLOS ONE. After careful consideration, we feel that it has merit but does not fully meet PLOS ONE’s publication criteria as it currently stands. Therefore, we invite you to submit a revised version of the manuscript that addresses the points raised during the review process.

We look forward to receiving your revised manuscript.

Kind regards,

Rami Salim Najjar, Ph.D.

Academic Editor

PLOS ONE

Journal Requirements:

2. We note that you have selected “Clinical Trial” as your article type. PLOS ONE requires that all clinical trials are registered in an appropriate registry (the WHO list of approved registries is at https://www.who.int/clinical-trials-registry-platform/network/primary-registries " https://www.who.int/clinical-trials-registry-platform/network/primary-registries and more information on trial registration is at http://www.icmje.org/about-icmje/faqs/clinical-trials-registration/ ). Please state the name of the registry and the registration number (e.g. ISRCTN or ClinicalTrials.gov ) in the submission data and on the title page of your manuscript. a) Please provide the complete date range for participant recruitment and follow-up in the methods section of your manuscript. b) If you have not yet registered your trial in an appropriate registry, we now require you to do so and will need confirmation of the trial registry number before we can pass your paper to the next stage of review. Please include in the Methods section of your paper your reasons for not registering this study before enrolment of participants started. Please confirm that all related trials are registered by stating: “The authors confirm that all ongoing and related trials for this drug/intervention are registered”. Please see http://journals.plos.org/plosone/s/submission-guidelines#loc-clinical-trials for our policies on clinical trials.

4. Please remove all personal information, ensure that the data shared are in accordance with participant consent, and re-upload a fully anonymized data set.

Reviewers' comments:

Reviewer's Responses to Questions

**Comments to the Author**

1. Is the manuscript technically sound, and do the data support the conclusions?

Reviewer #1: Yes

Reviewer #2: Yes

2. Has the statistical analysis been performed appropriately and rigorously? 

Reviewer #1: No

Reviewer #2: Yes

3. Have the authors made all data underlying the findings in their manuscript fully available?

Reviewer #1: Yes

Reviewer #2: Yes

4. Is the manuscript presented in an intelligible fashion and written in standard English?

Reviewer #1: Yes

Reviewer #2: Yes

5. Review Comments to the Author

Reviewer #1: The paper investigates the effects of boysenberry juice (BoyJ) containing boysenberry anthocyanins (BoyACs) on postprandial energy metabolism, particularly focusing on diet-induced thermogenesis (DIT) and fat oxidation in healthy adults.

Below are some comments/questions:

1. Sample Size Justification: The target sample size (n=22) was based on a pilot study measuring cold-induced BAT activation, but this study assessed DIT under thermoneutral conditions. Was the effect size for DIT expected to be similar to cold-exposure BAT effects? If not, how was the power calculation adjusted? In addition, COVID-19 reduced the final sample to n=19 (14% dropout). Was an intention-to-treat (ITT) analysis considered to mitigate bias from dropouts?

2. The significant fat oxidation finding contrasts with null DIT/BAT results. Could non-BAT mechanisms (e.g., hepatic or muscle metabolism) explain this effect? Were other pathways (e.g., AMPK/SIRT1) explored?

3. ΔTscv–ch P-value Discrepancy: Text states p=0.210 for ΔTscv–ch (BAT temperature), but Table 4 reports p=0.133. Which statistical test was primary (mixed-model vs. t-test)? Please clarify.

4. Fat oxidation is reported as mg/min in tables but g/min in supplemental files (S2/S4). Can units be standardized to avoid confusion?

5. Washout Period: A 4-week washout was used, but no pharmacokinetic data confirmed BoyAC clearance. Was this duration validated for anthocyanins or their metabolic effects?

6. BAT Measurement Sensitivity: BAT activity was assessed via skin temperature (not gold-standard PET/CT). Could the null BAT result reflect low sensitivity of thermography under thermoneutral conditions?

7. Post Hoc Power: The observed power for DIT was ~30% (when small effect size: d=0.35 is used). Should the DIT result be framed as "inconclusive due to underpowering" rather than "no effect"?

8. Participants were healthy adults (BMI 18.5–25). Would the findings extend to metabolic syndrome or obese populations?

Reviewer #2: Thank you for the invitation to review this impressive randomized controlled crossover human clinical trial (with washout period!) investigating the potential for boysenberries to modulate energy expenditure and substrate utilization. It takes a village of scientists to perform this careful gold standard in our field of nutritional sciences. This is a strong backbone of empirical research – my suggestions are reserved simply for adding some details where I thought it would help the authors tell their story more effectively, so as to be better received by interested readers.

Compliments:

I perform the same type of statistical testing – it is honest, and is also sensitive to detect changes that effectively tease apart subtle diet effects on EE, substrate metabolism, or both. Cheers to the research team for an excellent design.

19 participants is a robust and hard-earned pilot dataset – as the authors pointed out they have achieved statistical power – I empathize with the mentality to tell the readers this project was upended by the pandemic – my gentle point is you have power and you have discovered meaningful differences. Beyond what is stated re the CONSORT on line 98, I propose omitting mention of the hardships of the pandemic (lines 181-183, and lines 201-202, for example) unless there is additional meaningful context I missed – in which case I meant no offense.

Minor suggestions – this is mostly commentary for your team to consider rather than pointed criticisms or “required” corrections:

Line 68 – I suggest replacing “fat consumption” with “fat oxidation”

Lines 70-75 – great tie in with the metabolic pathway activation – consider briefly elaborating on SIRT1 in relation to PGC1a and AMPK – one proposed axis of anthocyanin activation.

Line 111 – Observing meaningful effects with 61 mg of anthocyanins is remarkable – more below on that topic.

Line 122 – Simple randomization was used – please consider reporting how this affected your sex balance across sequences – I suggest using “covariate adaptive randomization” (stratify sequence on gender) next time.

Line 194-195: I again commend the excellent approach to statistical testing. In one of our previous studies, we noted possible carryover effects when there was only a 1-week crossover (being driven by 1 volunteer in particular, who was also technically insulin resistant). We therefore became more conservative in follow-up studies and increased our washout period to 2 weeks. We only extended to 3 weeks in a recent study due to metabolic cart limitations. My suspicion is more “metabolically compromised” study volunteers (BMI > 25 and with associated metabolic stress) may heighten risk of carryover effects. In essence, I suggest your research team consider shortening your washout period, if time constraints of your current washout period length are complicating your productivity.

Lines 222-225 – it is interesting that you detected a significant difference in fat oxidation but not the RQ. In our hands, any time we observed significant effects on fat oxidation, the RQ also typically was significantly different (rationale since the fat oxidation calculation incorporates VO2 and VCO2). I wonder if your significance in one but not both is also a testament to more subtle differences. Considering the relatively low daily dose of anthocyanins (we hypothesize there is a threshold dose to reach measurable differences, at least in overweight/obese volunteers).

Lines 238-278 – very nice discussion. Mentioned above, the two things that impress me with your work are the relatively low dose of daily anthocyanins, and that you tested in very healthy individuals (by BMI standards, at least, indicating your cohort was within the healthy weight range). You may consider talking to the potency of boysenberry juice, as I do not recollect a study that could demonstrate detectable changes at 60 mg daily dose – this may speak to the potency of the anthocyanin composition of your berry juice – it could also be a testament to threshold doses for different body sizes? Even with the interesting nuance – I caution you to consider higher doses if you intend to test overweight/obese volunteers in future work. We have fed up to 720 mg of elderberry juice for 7 days – despite the low palatability of 100% juice, only 1 volunteer mentioned upset stomach.

289-290 – strong conclusion – well done.

Best wishes in your future pursuits on this topic – thank you for the chance to help elevate your work.

6. PLOS authors have the option to publish the peer review history of their article (what does this mean? ). If published, this will include your full peer review and any attached files.

**Do you want your identity to be public for this peer review?** For information about this choice, including consent withdrawal, please see our Privacy Policy .

Reviewer #1: No

Reviewer #2: **Yes: ** Patrick Solverson

---

## [Author Response · Author response to Decision Letter 1]

9 Jul 2025

Responses to the Comments by the Editors:

1. Please ensure that your manuscript meets PLOS ONE's style requirements, including those for file naming. The PLOS ONE style templates can be found at https://journals.plos.org/plosone/s/file?id=wjVg/PLOSOne_formatting_sample_main_body.pdf　and　https://journals.plos.org/plosone/s/file?id=ba62/PLOSOne_formatting_sample_title_authors_affiliations.pdf

Reply:

Thank you for providing the format. The format has been corrected to PLOS ONE style.

2. We note that you have selected “Clinical Trial” as your article type. PLOS ONE requires that all clinical trials are registered in an appropriate registry (the WHO list of approved registries is at https://www.who.int/clinical-trials-registry-platform/network/primary-registries" https://www.who.int/clinical-trials-registry-platform/network/primary-registries and more information on trial registration is at http://www.icmje.org/about-icmje/faqs/clinical-trials-registration/). Please state the name of the registry and the registration number (e.g. ISRCTN or ClinicalTrials.gov) in the submission data and on the title page of your manuscript. a) Please provide the complete date range for participant recruitment and follow-up in the methods section of your manuscript. b) If you have not yet registered your trial in an appropriate registry, we now require you to do so and will need confirmation of the trial registry number before we can pass your paper to the next stage of review. Please include in the Methods section of your paper your reasons for not registering this study before enrolment of participants started. Please confirm that all related trials are registered by stating: “The authors confirm that all ongoing and related trials for this drug/intervention are registered”. Please see http://journals.plos.org/plosone/s/submission-guidelines#loc-clinical-trials for our policies on clinical trials.

Reply:

Thank you for providing the necessary information for clinical trial articles.

The registration number has been added to the abstract.

Line 35:

(registration number: UMIN000047413)

The intervention period was also specified in the methods.

Line 89:

The trial was initiated on 4/6/2022 and completed on 3/10/2022, during which participants received the intervention.

Responses to the Comments by the Reviewer 1:

The paper investigates the effects of boysenberry juice (BoyJ) containing boysenberry anthocyanins (BoyACs) on postprandial energy metabolism, particularly focusing on diet-induced thermogenesis (DIT) and fat oxidation in healthy adults.

Reply:

Thank you for reviewing our paper and for accurately understanding the main focus of our research. We are especially grateful for your valuable comments regarding the statistical aspects, which helped us improve the quality of the manuscript.

1. Sample Size Justification: The target sample size (n=22) was based on a pilot study measuring cold-induced BAT activation, but this study assessed DIT under thermoneutral conditions. Was the effect size for DIT expected to be similar to cold-exposure BAT effects? If not, how was the power calculation adjusted?

Reply:

In the study published in the International Journal of Obesity (volume 45, pages 2499–2505, 2021), 21 male participants were evaluated for DIT and 23 male participants for cold induced thermogenesis (CIT), with both groups showing a significant increase in high BAT activity. This suggests that similar effects may be expected between DIT and CIT. However, due to the differences in the metrics, the analysis was conducted with a power calculation of 95%.

The original was revised as follows.

Line 192:

In previous literature(15), 21 male participants were evaluated for DIT and 23 male participants for cold induced thermogenesis (CIT) CIT, with both groups showing a significant increase in high BAT activity. This suggests that similar effects may be expected between DIT and CIT. however, due to the differences in the metrics, the analysis was conducted with a power calculation of 95%.

In addition, COVID-19 reduced the final sample to n=19 (14% dropout). Was an intention-to-treat (ITT) analysis considered to mitigate bias from dropouts?

Reply:

Thank you for your question regarding dropouts. Participants who dropped out did not obtained any outcome data after randomization, so they were excluded from the analysis, and we performed Full Analysis Set analysis. We did not specify the reason for excluding the participants from the analysis, which may have resulted in some ambiguity. This point is made clear in the revised manuscript.

Line 99

As a result, 19 participants completed the study. Three dropouts were excluded from the statistical analysis because all outcome data were not obtained, and the Full Analysis Set was performed on 19 participants.

2. The significant fat oxidation finding contrasts with null DIT/BAT results. Could non-BAT mechanisms (e.g., hepatic or muscle metabolism) explain this effect? Were other pathways (e.g., AMPK/SIRT1) explored?

Reply:

Thank you for your valuable suggestions. We have added the following to our considerations.

Line 362:

AMP-activated protein kinase (AMPK) and sirtuin 1, as metabolic sensors, regulate the activity of the master regulator of mitochondria, Peroxisome proliferator-activated receptor gamma coactivator-1-alpha (PGC-1α), and they interact to control energy metabolism in the liver, muscle, and adipose tissue, playing a crucial role in maintaining energy homeostasis(25). Furthermore, it has been reported that the stimulation of sirtuin 1, AMPK, and PGC-1α, along with their antioxidant effects, enables anthocyanins to promote mitochondrial density and function in skeletal muscle, liver, and adipose tissue(26). BoyAC has been suggested in preclinical studies to indirectly maintain BAT activation by preserving vascular homeostasis through sirtuin 1(8). This effect may influence not only BAT but also a wide range of organs.

3. ΔTscv–ch P-value Discrepancy: Text states p=0.210 for ΔTscv–ch (BAT temperature), but Table 4 reports p=0.133. Which statistical test was primary (mixed-model vs. t-test)? Please clarify.

Reply:

Thank you for pointing out the discrepancy. The correct p-value for ΔTscv–ch (BAT temperature) is indeed p=0.133. We have corrected the text and the p-value in Fig 3C to reflect this.

Line 248:

There was no significant difference in ΔTscv−ch after the meal between the placebo and BoyJ groups (Fig 3C, p = 0.133), as determined by linear mixed-effects models.

4. Fat oxidation is reported as mg/min in tables but g/min in supplemental files (S2/S4). Can units be standardized to avoid confusion?

Reply:

Thank you for your helpful comment. We have standardized the units in the supplemental files (S2/S4) to mg/min to avoid any confusion.

5. Washout Period: A 4-week washout was used, but no pharmacokinetic data confirmed BoyAC clearance. Was this duration validated for anthocyanins or their metabolic effects?

Reply:

We believe that the 4-week washout period is sufficient. As evidence, we statistically analyzed the carryover effects and cited literature demonstrating the metabolism of anthocyanins, which we have added to the discussion.

Line 210:

To evaluate the potential carryover effect, the sum of DIT (%) for Test 1 and Test 2 was calculated, and the sequences 1 and 2 were compared using a Student's t-test.

Line 251:

Additionally, no carryover effect was observed for DIT (p=0.595).

Line 310:

A previous study(18) have provided evidence that the consumption of the anthocyanin cyanidin-3-glucoside results in the detection of anthocyanin-derived metabolites in humans for up to 48 hours. They reported that the half-life of these metabolites, including other components, ranged from 12.44 ± 4.22 hours to 51.62 ± 22.55 hours. Therefore, it is considered that the carryover of metabolites after four weeks has minimal impact. Additionally, as the carryover effect was statistically negated in this study, a 4-week washout period was likely sufficient.

6. BAT Measurement Sensitivity: BAT activity was assessed via skin temperature (not gold-standard PET/CT). Could the null BAT result reflect low sensitivity of thermography under thermoneutral conditions?

Reply:

We agree that the method used for assessing BAT activity via skin temperature is typically employed during cold-loading conditions. We believe that the null BAT result may reflect low sensitivity of thermography under thermoneutral conditions, highlighting the limitations of the method.

7. Post Hoc Power: The observed power for DIT was ~30% (when small effect size: d=0.35 is used). Should the DIT result be framed as "inconclusive due to underpowering" rather than "no effect"?

Reply:

We recognize that there were issues with sample size for DIT, and we agree that framing the results as "inconclusive due to underpowering" is more appropriate. We have added this consideration to the discussion and revised the conclusion based on your suggestions.

Line 323:

In our analysis, the post hoc power calculation yielded a value of 0.250 for DIT, indicating that the study was underpowered to detect an effect of the indicated size. The calculated effect size d was 0.311, suggesting a small effect was present. However, the low power highlights that the results may be considered inconclusive due to underpowering. Therefore, we recommend that future studies with larger sample sizes be conducted to more definitively assess the impact.

Line 382:

Although no significant changes were observed in DIT or thermogenesis in the BAT area, the low power of the study limited our ability to draw definitive conclusions regarding DIT.

8. Participants were healthy adults (BMI 18.5–25). Would the findings extend to metabolic syndrome or obese populations?

Reply:

This study focused on healthy adults with a BMI of 18.5–25, and therefore, the findings do not apply to populations with metabolic syndrome or obesity. We acknowledge that future research is needed to investigate the effects in more unhealthy subjects.

Responses to the Comments by the Reviewer 2:

Thank you for the invitation to review this impressive randomized controlled crossover human clinical trial (with washout period!) investigating the potential for boysenberries to modulate energy expenditure and substrate utilization. It takes a village of scientists to perform this careful gold standard in our field of nutritional sciences. This is a strong backbone of empirical research – my suggestions are reserved simply for adding some details where I thought it would help the authors tell their story more effectively, so as to be better received by interested readers.

I perform the same type of statistical testing – it is honest, and is also sensitive to detect changes that effectively tease apart subtle diet effects on EE, substrate metabolism, or both. Cheers to the research team for an excellent design.

Reply:

Thank you for your insightful comments and for understanding our research. We appreciate your feedback and aim to utilize your suggestions to improve our manuscript and make it more effective for our readers.

19 participants is a robust and hard-earned pilot dataset – as the authors pointed out they have achieved statistical power – I empathize with the mentality to tell the readers this project was upended by the pandemic – my gentle point is you have power and you have discovered meaningful differences. Beyond what is stated re the CONSORT on line 98, I propose omitting mention of the hardships of the pandemic (lines 181-183, and lines 201-202, for example) unless there is additional meaningful context I missed – in which case I meant no offense.

Reply:

Thank you for your valuable feedback. You are correct that we may have overemphasized the impact of COVID-19. Following your suggestion, we will remove references to COVID-19 from the manuscript to improve clarity and focus.

Line 201: omiited

Nonetheless, owing to the impact of the COVID-19 pandemic, the target sample size was not attained.

Line 222: modified

During the trial period, one participant dropped out before randomization, and three participants dropped out after randomization, which hindered our ability to collect the target sample size.

Line 354: omiited

The reduced sample size due to the impact of the COVID-19 pandemic (21) may have resulted in inadequate statistical power, potentially reducing the ability to detect changes in these outcomes.

Minor suggestions – this is mostly commentary for your team to consider rather than pointed criticisms or “required” corrections:

Line 68 – I suggest replacing “fat consumption” with “fat oxidation”

Thank you for your suggestion. I have made the correction by replacing "fat consumption" with "fat oxidation" as recommended.

Line 66:

leading to increased fat oxidation and EE

Lines 70-75 – great tie in with the metabolic pathway activation – consider briefly elaborating on SIRT1 in relation to PGC1a and AMPK – one proposed axis of anthocyanin activation.

Reply:

Thank you for your valuable suggestions. We have added the following to our considerations.

Line 362:

AMP-activated protein kinase　 (AMPK) and sirtuin 1, as metabolic sensors, regulate the activity of the master regulator of mitochondria, Peroxisome proliferator-activated receptor gamma coactivator-1-alpha (PGC-1α), and they interact to control energy metabolism in the liver, muscle, and adipose tissue, playing a crucial role in maintaining energy homeostasis(25). Furthermore, it has been reported that the stimulation of sirtuin 1, AMPK, and PGC-1α, along with their antioxidant effects, enables anthocyanins to promote mitochondrial density and function in skeletal muscle, liver, and adipose tissue(26). BoyAC has been suggested in preclinical studies to indirectly maintain BAT activation by preserving vascular homeostasis through sirtuin 1(8). This effect may influence not only BAT but also a wide range of organs.

Line 122 – Simple randomization was used – please consider reporting how this affected your sex balance across sequences – I suggest using “covariate adaptive randomization” (stratify sequence on gender) next time.

We confirmed that there was no significant difference in the number of males and females, with a p-value of 0.515 from the chi-squared test. The sample size and analysis results have been added below. Additionally, we would like to consider "covariate-adaptive randomization" for future trials.

Line 212:

To consider the sex balance across sequences, we analyzed the effects using the chi-squared test.

Line 224

We confirmed that there was no significant difference in the number of males and females across sequences (Sequence 1: 7 males and 3 females; Sequence 2: 5 males and 4 females), with a p-value of 0.515.

Line 194-195: I again commend the excellent approach to statistical testing. In one of our previous studies, we noted possible carryover effects when there was only a 1-week crossover (being driven by 1 volunteer in particular, who was also technically insulin resistant). We therefore became more conservative in follow-up studies and increased our washout period to 2 weeks. We only extended to 3 weeks in a recent study due to metabolic cart limitations. My suspicion is more “metabolically compromised” study volunteers (BMI > 25 and with associated metabolic stress) may heighten risk of carryover effects. In essence, I suggest your research team consider shortening your washout period, if time constraints of your current washout period length are complicating your productivity.

Reply:

As you pointed out, the duration of the study posed several constraints in terms of participant recruitment and evaluation. Therefore, your comments are very helpful for guiding future trials. Considering the half-lives of metabolites, it can be estimated that a washout period of 2 to 3 weeks could be sufficient to achieve the desired effects, and we will take this into account in future studies.

Lines 222-225 – it is interesting that you detected a significant difference in fat oxidation but not the RQ. In our hands, any time we observed significant effects on fat oxidation, the RQ also typically was significantly different (rationale since t

---

## [Decision Letter · Decision Letter 1]

5 Aug 2025

Effects of boysenberry on postprandial energy metabolism in healthy adults:

A randomized controlled crossover trial

PONE-D-25-23279R1

Dear Dr. Furuuchi,

We’re pleased to inform you that your manuscript has been judged scientifically suitable for publication and will be formally accepted for publication once it meets all outstanding technical requirements.

Kind regards,

Rami Salim Najjar, Ph.D.

Academic Editor

PLOS ONE

Additional Editor Comments (optional):

Reviewers' comments:

Reviewer's Responses to Questions

**Comments to the Author**

1. If the authors have adequately addressed your comments raised in a previous round of review and you feel that this manuscript is now acceptable for publication, you may indicate that here to bypass the “Comments to the Author” section, enter your conflict of interest statement in the “Confidential to Editor” section, and submit your "Accept" recommendation.

Reviewer #1: All comments have been addressed

2. Is the manuscript technically sound, and do the data support the conclusions?

Reviewer #1: (No Response)

3. Has the statistical analysis been performed appropriately and rigorously? 

Reviewer #1: (No Response)

4. Have the authors made all data underlying the findings in their manuscript fully available?

Reviewer #1: (No Response)

5. Is the manuscript presented in an intelligible fashion and written in standard English?

Reviewer #1: (No Response)

6. Review Comments to the Author

Reviewer #1: (No Response)

7. PLOS authors have the option to publish the peer review history of their article (what does this mean? ). If published, this will include your full peer review and any attached files.

**Do you want your identity to be public for this peer review?** For information about this choice, including consent withdrawal, please see our Privacy Policy .

Reviewer #1: No

---

## [Editor Report · Acceptance letter]

PONE-D-25-23279R1

PLOS ONE

Dear Dr. Furuuchi,

I'm pleased to inform you that your manuscript has been deemed suitable for publication in PLOS ONE. Congratulations! Your manuscript is now being handed over to our production team.

Kind regards,

on behalf of

Dr. Rami Salim Najjar

Academic Editor

PLOS ONE